# Emotional Intelligence Measures: A Systematic Review

**DOI:** 10.3390/healthcare9121696

**Published:** 2021-12-07

**Authors:** Lluna María Bru-Luna, Manuel Martí-Vilar, César Merino-Soto, José L. Cervera-Santiago

**Affiliations:** 1Department of Basic Psychology, Faculty of Psychology and Speech Therapy, Universitat de València, 46010 Valencia, Spain; llunamaria.bl@gmail.com; 2Psychology Research Institute, Universidad de San Martín de Porres, Lima 15102, Peru; 3Department of Psychology, Faculty of Psychology, Universidad Nacional Federico Villarreal, San Miguel 15088, Peru; jcervera@unfv.edu.pe

**Keywords:** emotional intelligence, systematic review, test, measure, questionnaire, scale

## Abstract

Emotional intelligence (EI) refers to the ability to perceive, express, understand, and manage emotions. Current research indicates that it may protect against the emotional burden experienced in certain professions. This article aims to provide an updated systematic review of existing instruments to assess EI in professionals, focusing on the description of their characteristics as well as their psychometric properties (reliability and validity). A literature search was conducted in Web of Science (WoS). A total of 2761 items met the eligibility criteria, from which a total of 40 different instruments were extracted and analysed. Most were based on three main models (i.e., skill-based, trait-based, and mixed), which differ in the way they conceptualize and measure EI. All have been shown to have advantages and disadvantages inherent to the type of tool. The instruments reported in the largest number of studies are Emotional Quotient Inventory (EQ-i), Schutte Self Report-Inventory (SSRI), Mayer-Salovey-Caruso Emotional Intelligence Test 2.0 (MSCEIT 2.0), Trait Meta-Mood Scale (TMMS), Wong and Law’s Emotional Intelligence Scale (WLEIS), and Trait Emotional Intelligence Questionnaire (TEIQue). The main measure of the estimated reliability has been internal consistency, and the construction of EI measures was predominantly based on linear modelling or classical test theory. The study has limitations: we only searched a single database, the impossibility of estimating inter-rater reliability, and non-compliance with some items required by PRISMA.

## 1. Introduction

### 1.1. Emotional Intelligence

Emotional intelligence (EI) was first described and conceptualized by Salovey and Mayer [1] as an ability-based construct analogous to general intelligence. They argued that individuals with a high level of EI had certain skills related to the evaluation and regulation of emotions and that consequently they were able to regulate emotions in themselves and in others in order to achieve a variety of adaptive outcomes. This construct has received increasing attention from both the scientific community and the general public due to its theoretical and practical implications for daily life. The same authors defined EI as “the ability to carry out accurate reasoning about emotions and the ability to use emotions and emotional knowledge to enhance thought” [2] (p. 511). This definition suggests that EI is far from being conceptualized as a one-dimensional attribute and that a multidimensional operationalization would be theoretically coherent.

### 1.2. Conceptualizations of Emotional Intelligence

However, over the past three decades, different ways of conceptualizing EI have emerged, which are mainly summarized in three models: ability, trait, and mixed. These models have influenced the construction of measuring instruments. In the ability model, developed by Mayer and Salovey, EI is seen as a form of innate intelligence made up of several capacities that influence how people understand and manage their own emotions and those of others. These emotion processing skills are: (1) perception, evaluation and expression of emotions, (2) emotional facilitation of thought, (3) understanding and analysis of emotions, and (4) reflective regulation of emotions [3,4]. Consistent with this conceptualization, the measures were designed as performance tests. Subsequently, the model proposed by Petrides and Furnham [5], the trait model, was developed. This model defines EI as a trait; that is, as a persistent behaviour pattern over time (as opposed to skill, which increases with time and training), and it is associated with dispositional tendencies, personality traits or self-efficacy beliefs. It is composed of fifteen personality dimensions, grouped under four factors: well-being, self-control, emotionality and sociability [6]. The last of the three main models of conceptualization of EI is the mixed one. It is made up of two large branches that consider this construct a mixture of traits, competencies and abilities. According to the first one, developed by Bar-On [7], EI is a set of non-cognitive abilities and competences that influence the ability to be successful in coping with environmental demands and pressures, and it is composed of five key components: intrapersonal skills, interpersonal skills, adaptation skills, stress management skills and general mood. The second one, proposed by Goleman [8], also conceptualizes EI as a mixed model that shares certain aspects with the Bar-On model. It is made up of the following elements: recognition of one’s own emotions, management of emotions, self-motivation, recognition of emotions in others, and management of relationships. These emotional and social competencies would contribute to managerial performance and leadership.

### 1.3. Importance of Emotional Intelligence

To date, the importance that academics attach to the study of EI has been recognized by the literature in many areas, such as the workplace. For example, in professions where working with people is needed, burnout syndrome is common. It is a syndrome that is expressed by an increase in emotional exhaustion and indifference, as well as by a decrease in professional effectiveness [9]. To date, numerous studies have shown that EI can help change employee attitudes and behaviours in jobs involving emotional demands by increasing job satisfaction and reducing job stress [10,11,12,13]. Likewise, on the one hand, it has been found that certain psychological variables, including EI and social competence, are related to less psychological distress. On the other hand, the acquisition of emotional and social skills can serve to develop resilience, which is a protective variable against psychological distress [14].

### 1.4. Types of Measures

With the challenge of choosing the conceptual model of EI also appears the challenge of choosing the appropriate measures to estimate it. For this reason, part of the work developed in the field of EI has focused on the creation of objective instruments to evaluate aspects associated with this construct. Most of them have been created around the main conceptualization models described in the previous paragraphs. Ability-based tools indicate people’s ability to understand emotions and how they work. These types of tests require participants to solve problems that are related to emotions and that contain answers deemed correct or incorrect (e.g., participants see several faces and respond by indicating the degree to which a specific emotion is present in the face). These instruments are maximal capacity tests and, unlike trait tests, they are not designed to predict typical behaviour. Ability EI instruments are usually employed in situations where a good theoretical understanding of emotions is required [15].

Trait-based instruments are generally composed of self-reported measures and are often developed as scales where there are no correct or incorrect answers, but the individual responds by choosing the item which relates more or less to their behaviour (e.g., “Understanding the needs and desires of others is not a problem for me”). They tend to measure typical behaviour, so they tend to provide a good prediction of actual behaviours in various situations [5]. Trait EI is a good predictor of effective coping styles when facing everyday stressors, both in adults and children, so these instruments are often used in situations characterized by stressors such as educational and employment contexts [15].

Questionnaires based on the EI mixed conceptualization often measure a combination of traits, social skills, competencies, and personality measures through self-reported modality (e.g., “When I am angry with others, I can tell them”). Some measures typically take 360-degree forms of assessment too (i.e., a self-report along with reports from supervisors, colleagues and subordinates). They are generally used in work environments, since they are often designed to predict and improve workplace performance and are often focused on emotional competencies that correlate with professional success. Despite the different ways of conceptualizing EI, there are some conceptual similarities between most instruments: they are hierarchical (i.e., they produce a total EI score along with scores on the different dimensions) and they have several conceptual overlaps that often include emotional perception, emotional regulation, and adaptive use of emotions [15].

### 1.5. Relevance of the Study

The proliferation of EI measures has received a lot of attention. However, this has not been the case in studies that synthesize their psychometric qualities, as well as those that describe their strengths and limitations. Therefore, there is a lack of studies that collect, with a wide review coverage, the instruments developed in recent years. The few reviews that can be found [16,17,18,19] are limited to describing both the most popular measures (e.g., Mayer–Salovey–Caruso Emotional Intelligence Test [MSCEIT], Emotional Quotient Inventory [EQ-i], Trait Meta-Mood Scale [TMMS], Trait Emotional Intelligence Questionnaire [TEIQue], or Schutte Self-Report Inventory [SSRI]) and those validated only in English, producing an apparent “Tower of Babel” effect (i.e., the over-representation of studies in one language and the under-representation in others) [20]. This is a problem that is not only more common than is believed, but it is also persistent [21]. This effect produces a barrier for the complete knowledge of current EI measures, the breadth of their uses in different contexts, and their incorporation into substantive studies relevant to multicultural understanding. In summary, it reduces the commonality of efforts made in different contexts to identify common and communicable objectives [22], specifically around the study of EI.

Therefore, a systematic review allows us to establish a knowledge base that contributes by (a) guiding and developing research efforts, (b) assisting in professional practice when choosing the most appropriate model in possible practical scenarios, and (c) facilitating the design of subsequent systematic evaluative reviews and meta-analysis of relevant psychometric parameters (e.g., factorial loads, reliability coefficients, correlations, etc.). For this reason, the aim of this article is to provide an updated systematic review of the existing instruments that allow the evaluation of EI in professionals, focusing on the description of its characteristics, as well as on its psychometric properties (reliability and validity). This systematic review is characterized by having a wide coverage (i.e., studies published in languages other than English) and having as a framework a consensus of description and taxonomy of valid evidence (i.e., “Standards”) [23].

## 2. Materials and Methods

This work contains a systematic review of the scientific literature published to date that includes measurements of EI. For its preparation, the guidelines proposed in the PRISMA statement [24] (Table A1) carrying out systematic reviews have been followed. Regarding the evaluation of the quality of the articles, since our study does not analyse the studies that employ the EI instruments but the instruments themselves, the assessment of the internal or external validity of the studies is not applicable to this research. However, an internationally proposed guide to the study of the validity of instruments, called “Standards”, has also been used [23]. It presents guidelines for the study of the composition, use, and interpretation of what a test aims to measure and proposes five sources of validity of evidence: content, response processes, internal structure, relationship with other variables and the consequences of testing. Likewise, a recently proposed registration protocol [25] for carrying out systematic reviews has also been followed based on the five validity sources of the “Standards”.

### 2.1. Information Sources

The bibliographic search was carried out in three phases: an initial search to obtain an overview of the current situation, a system that applies inclusion–exclusion criteria, and a manual search to evaluate the results obtained. The search was conducted in February 2021 in the Web of Science (WoS) database, including all articles published from 1900 to 2020 (inclusive). This database was selected to perform the search because (a) it is among the databases that allows for a more efficient and adequate search coverage [26]; (b) it provides a better quality of indexing and of bibliographic records in terms of accuracy, control and granularity of information compared to other databases [27]; (c) the results are highly correlated with those of other search engines (e.g., Embase, MEDLINE and Google Scholar) [26]; (d) it is controlled by a human team specialising in the selection of its content (i.e., it is not fully automated) [28]; and (e) it has experienced a constant increase in scientific publications [29].

### 2.2. Eligibility Criteria

Although no protocol was written or registered prior to the research, the inclusion and exclusion criteria for articles and instruments were previously defined. The search was conducted according to these criteria.

#### 2.2.1. Inclusion Criteria

The inclusion criteria for the studies are made up of the following points: (a) published in peer-reviewed journals, (b) presented as full articles or short communications, (c) containing empirical and quantifiable results on psychometric properties (i.e., not only narrative descriptions), (d) containing cross-sectional or longitudinal designs, (e) written in any language (in order to collect as many instruments as possible, as well as to reduce the “Tower of Babel” effect) [20], and (f) published from 1900 to 2020 (to maximize the identification of EI measures).

As for the inclusion criteria of the instruments, they are made up of the following points: (a) instruments that measure EI, (b) articles that are the first creation study of the instrument, (c) instruments aimed at people over 18 years, (d) instruments that can be applied in the workplace.

#### 2.2.2. Exclusion Criteria

On the other hand, research that presented at least one of the following exclusion criteria was discarded: (a) contains synthesis studies (i.e., systematic reviews or meta-analyses), instrument manuals or narrative articles of instrument characteristics, (b) contains only qualitative research designs, (c) published after 2020.

Instruments that presented at least one of the following exclusion criteria were discarded: (a) instruments that were validations of the original one, (b) instruments aimed at people under 18, (c) instruments to be used in areas specifically different from the workplace.

### 2.3. Search Strategy

All available methods to obtain empirical answers have been included so as to maximize the coverage of the results. The following terms were included: test, measure, questionnaire, scale and instrument. The combinations of terms used were: “emotional intelligence AND test”, “emotional intelligence AND measure”, “emotional intelligence AND questionnaire”, “emotional intelligence AND scale”, and “emotional intelligence AND instrument”. Only those article-type studies were selected.

In the selection process, the title, abstract and keywords of the studies identified in the search were reviewed with the aforementioned criteria. This was carried out by only one of the authors.

### 2.4. Data Collection

The data to be extracted from each of the instruments were also defined in advance, ensuring that the information was extracted in a uniform manner. The selected documents were then recorded in a Microsoft Excel spreadsheet to check for duplicate records.

Thus, the name of the instrument and its acronym, the language and country in which it was created, and its structural characteristics (i.e., type of measurement, number of items, dimensions and items of which they were composed, and theoretical model) were extracted together with relevant psychometric information (i.e., reliability and validity). This procedure was also carried out by the same author. Articles that used different versions of the original EI instrument were accepted, but the analysis was made only on their originals. Instruments whose original manuscript were inaccessible were discarded (*n* = 10), but they are presented at the end of the results. All those articles that were duplicated or that had used measures aimed at people under 18 or for contexts specifically different from the professional area (e.g., school contexts, sports contexts, etc.) were eliminated. The search process and the number of selected and excluded results can be seen in Figure 1. Regarding the ethical standards, no ethical approval or participant consent is required for this type of research (i.e., systematic review).

## 3. Results

A total of 40 instruments were found (Table 1 shows a synthesis of all of them). Below, a brief description of each one is presented, following which a division according to the theoretical model they use (i.e., ability-based model, trait-based model, mixed approach model, and others that do not correspond to any of them), and the psychometric properties of each one are explained.

### 3.1. Ability-Based Measures

The first category includes those instruments based on the ability-based model, mainly on that of Mayer and Salovey [4]. The first instrument created under this conceptualization is the Trait Meta-Mood Scale (TMMS) [30], a self-report scale designed to assess people’s beliefs about their own emotional abilities. It measures three key aspects of perceived EI: attention to feelings, emotional clarity and repair of emotions. It presents a very good reliability [80] and convergent validity with various instruments, although the authors recommend the use of a later version of 30 items. It also presents a widely used 24-item version [31] that has been validated in many countries.

Three years later, the Schutte Self-Report Emotional Intelligence (SSRI) test was developed [33]. This questionnaire is answered through a five-point Likert scale and is composed of one factor that is divided into three categories: appraisal and expression of emotion in the self and others, regulation of emotion in the self and others and utilization of emotions in solving problems. It shows excellent internal consistency. It presents negative correlations with instruments that measure alexithymia, depression and impulsivity among others, which confirms its convergent validity. There is a modified version [34] and an abbreviated version [35], and it has been translated into many languages.

The Multifactor Emotional Intelligence Scale (MEIS) [37] is another tool developed by the authors that originally defined and conceptualized EI. The MEIS is a scale made up of 12 different tasks that contains 402 items and it has been translated into several languages. However, it has strong limitations such as its length and the low internal consistency offered by some of the tasks (e.g., “blends” and “progressions”; *α* = 0.49 and 0.51, respectively). These authors developed, years later, the Mayer–Salovey–Caruso Emotional Intelligence Test (MSCEIT) [38]. The items developed for the MEIS served as the starting point for the MSCEIT. This measure is composed of a five-point Likert scale and multiple response items with correct and incorrect options, which comprise eight tasks. Each of the four dimensions is assessed through two tasks. It presents an adequate internal consistency. It currently has a revised version by the same authors, and another validated in a young population. In addition, it has been translated into many languages. This instrument has detractors. Its convergent validity has been questioned since no correlation has been found between the emotional perception scale of MSCEIT and other emotional perception tests [81]. As can be seen in Table 1, the MSCEIT has two different approaches to construct the score (consensus score and expert score). In the case of EI, it is difficult to classify an answer as correct or incorrect, so if a person responds in a different way to the experts or the average, it might mean that they have low emotional capacity or present a different way of thinking [81].

In the same year, three more instruments based on this conceptualization were developed in different countries. The first one, the Profile of Emotional Intelligence (PIEMO) [40] is an inventory developed in Mexico. Their items consist of a statement that represents a paradigmatic behaviour trait of EI with true and false answers. It is composed of eight independent dimensions that together constitute a profile. Its internal consistency is excellent and its validity has been tested by a confirmatory factor analysis and expert consultations on the items.

The second instrument is Wong and Law’s Emotional Intelligence Scale (WLEIS) [41]. It was developed in China to measure EI in a brief way in leadership and management studies. It has an adequate internal consistency and has positive correlations with the TMMS and the EQ-i. Subsequent studies have shown its predictive validity in relation to life satisfaction, happiness or psychological well-being, and its criteria’s validity with respect to personal well-being. Measurement equivalence of scores in different ethnic and gender groups has also been tested [82]. It has been translated into a multitude of languages and it is currently one of the most widely used instruments.

The third instrument is the Workgroup Emotional Intelligence Profile-3 (WEIP-3) [43]. It is a scale designed in Australia as a self-report to measure the EI of people in work teams. It has very good internal consistency and presents correlations with several instruments that prove its convergent validity. The authors made a particularly interesting finding in their study. Teams that scored lower in the WEIP-3 performed at lower levels in their work than those with high EI. This instrument has a short version and has been translated into different languages.

The Multidimensional Emotional Intelligence Assessment (MEIA) [45] was developed in the USA. The authors state that the test takes only 20 min. It has very good internal consistency. Its validity has been tested in different ways. Content validity was tested by independent experts who considered each element as representative of its target scale. Convergent validity was tested by significant correlations between the scores and personality tests. Finally, the lack of correlation between the MEIA and theoretically unrelated personality tests proved the divergent validity. It has a version for the work context.

The Sojo and Steinkopf Emotional Intelligence Inventory—Revised version (IIESS-R) [47] was developed in Venezuela to measure the three dimensions that compose it. It presents 34 phrases that describe the reactions of people with high EI, as well as contrary behaviours. It has excellent internal consistency and its content has been validated through expert judgment. It shows correlations with some scales of similar instruments and its internal structure has been tested by exploratory analysis and PCA.

In the original article of the Emotional Intelligence Questionnaire (EmIn), created for the Russian population [46], its author proposes his own model of ability-based EI that differs in some aspects from that proposed by Mayer and Salovey. Accordingly, he designed a questionnaire to measure the participants’ beliefs about their emotional abilities under this model. It is composed of two dimensions answered using a 4-point Likert scale. Their scales have a good internal consistency, but their validity has not been tested beyond the factor analysis of its internal structure. Years later, this same author developed the Videotest of Emotion Recognition [59], an instrument that uses videos as stimuli. It was also designed in Russia to obtain precision indexes in the recognition of the types of emotions, as well as the sensitivity and intensity of the observed emotions. It has 15 scales that measure through a single item each of the emotions recorded by the instrument. Its internal consistency is good. It is correlated with MSCEIT and EmIn, which proves its convergent validity.

Another instrument based on the Mayer and Salovey model is the Self-Rated Emotional Intelligence Scale (SREIS) [49]. It was developed throughout three studies that used the MSCEIT as a comparison. The first one did not show a very high correlation between the scores of both tools. In the second one, only men’s MSCEIT scores correlated with perceived social competence after personality measures remained constant. Finally, in the third only MSCEIT predicted social competence, but only for males again. Internal consistency was also not consistent throughout the three studies, as the *α* yielded values were 0.84, 0.77, and 0.66, respectively. Its internal structure was tested by a confirmatory factor analysis and the content of each item was validated by the judgment of students familiar with the Mayer and Salovey model. It has been translated into several languages.

The Emotional Intelligence Self-Description Inventory (EISDI) [49] is also a short instrument, consisting of four dimensions designed to assess EI in the workplace. It has an excellent internal consistency. It presents correlations with instruments such as the WLEIS and the SREIS and a discriminant validity with the Big Five Personality. The same year, the Greek Emotional Intelligence Scale (GEIS) [51] was developed in Greek to assess four basic dimensions of EI. Its internal consistency is very good, as well as its test–retest value. Its internal structure was verified by a PCA, and its convergent and divergent validity were tested by a series of studies with 12 different instruments.

MacCann and Roberts [51] developed two instruments to assess EI according to the ability-based model: the Situational Test of Emotion Management (STEM) and the Situational Test of Emotional Understanding (STEU). Both are made up of three dimensions and a similar number of items. The first one measures the management of emotions such as anger, sadness and fear, and it can be administered in two formats: multiple choice response and rate-the-extent (i.e., test takers rate the appropriateness, strength, or extent of each alternative, rather than selecting the correct alternative). The STEU presents a series of situations about context-reduced, personal-life context, and workplace-context, which provoke a main emotion that is the correct answer to be chosen by the participant among other incorrect ones. Both instruments have similar internal consistency for the multiple response format, while for the rate-the-extent format it is much higher. Both present criteria and convergent validity and have an abbreviated version.

The Emotional Skills and Competence Questionnaire (ESCQ) [53] is an instrument developed in Croatia that measures EI through three basic dimensions using a five-point Likert scale. The subscales have a reliability that varies between good an excellent, and they correlate with other EI and personality instruments. The ESCQ has been translated into several languages.

The Audiovisual Test of Emotional Intelligence (AVEI) [55] is an Israeli instrument aimed at educational settings related to care-centred professions. Their items are developed from primary and secondary emotions, both positive and negative. Each one consists of short videos generated by researchers with training in psychology and visual arts. People should choose the correct answer among 10 alternatives and it takes between 12 and 18 min to be completed. It requires computers equipped with audio. The internal consistency was calculated using ICC coefficients. It has content validations through expert consultations on the items and criteria since it correlates with measures traditionally related to EI.

The Geneva Emotion Recognition Test (GERT) [57] is a German test composed of 14 scales. The stimuli are, as in the AVEI, short image and audio videos recorded by five men and five women of different ages. Thus, people must choose which of the 14 emotions is being expressed by the actors, with the responses labelled as correct or incorrect. The reliability of the test is considered excellent, and the ecological and construct validity of the instrument has been tested.

The Test of Emotional Intelligence (TIE) [58] is developed in Poland. It consists of the same four dimensions as the MSCEIT. After providing participants with different emotional problems, they should indicate which emotion is most likely to occur or choose the most appropriate action. The score is based on expert judgment. It has a very good internal consistency. It has convergent validity since it correlates with the SSEIT and has construct since women scored higher than men.

The Self-Perception of Emotional Intelligence Questionnaire (EIQ-SP) [60] is an instrument designed in Portugal and composed of the four dimensions belonging to the Mayer and Salovey’s ability-based model. Their scales have good internal consistency and are correlated with each other.

The Three-Branch Emotional Intelligence Rating Scale Assessment (TEIRA) [61] and the Three-Branch Emotional Intelligence Forced-Choice Assessment (TEIFA) [61] were developed in 2015. The first is made up of three scales and is answered by a six-point Likert scale. It presents internal consistency between good and excellent and convergent validity with STEU-B and STEM-B. On the other hand, TEIFA presents a format of forced choice in order to avoid the problem of social desirability in the rating scales. In this format, participants must choose among several positive statements and therefore they cannot simply rate themselves highly on everything (e.g., “Which one is more like you: I know why my emotions change or I manage my emotions well”). It consists of the same items and dimensions as the TEIRA. The study does not report the reliability of TEIFA, as the reliability of the forced-choice tests is artificially high. It presents convergent validity with the SSRI.

A year later, the North Dakota Emotional Abilities Test (NEAT) [62] was developed in the USA to assess the ability to perceive, understand and control emotions in the workplace. It contains items that describe scenarios of work environments, in which the person must rate the extent of certain emotions that the protagonist would experience in a certain situation. The internal consistency of its scales varies between good and excellent and its internal structure has been tested by a confirmatory factor analysis. In addition, the predictive validity of the instrument has also been tested.

The Inventory of Perceived Emotional Intelligence (IIEP) [63] was developed in Argentina. It measures different components of intrapersonal and interpersonal EI. This inventory is answered using a five-point Likert scale and it has reliable dimensions. Its content validity has been tested through consultations with judges to evaluate the items.

The last of the instruments in this category is the Emotional Intelligence Test (EIT) [65]. It was developed in Russia and has four dimensions that assess EI in the workplace. It has excellent internal consistency and convergent validity tested by correlations with the MSCEIT 2.0. No information regarding the items that compose it has been found.

### 3.2. Measures Based on the Mixed Model

The second category includes those instruments based on the mixed EI model, mainly the Bar-On model [7] and the Goleman model [8]. The first instrument of this model is the Emotional Quotient Inventory (EQ-i) [7]. Its author was the first to define EI as a mixed concept between ability and personality trait. It is a self-report measure of behaviour that provides an estimate of EI and social intelligence. Their items are composed of short sentences that are answered using a five-point Likert scale. It takes about 30 min to complete, so other shorter versions have been developed, as well as a 360-degree version and a version for young people. It has been translated into more than 30 languages. It has an internal consistency between good and very good and its construct validity has been tested by correlations with other variables.

Emotional Competence Inventory 2.0 (ECI 2.0) [67], also called ESCI, is a widely used instrument. It was developed in the USA by another of the authors who conceptualized the mixed model of EI. It was designed in a 360-degree version to assess the emotional competencies of individuals and organizations. The internal consistency of others’ ratings is good, while that of oneself is questionable, and it shows positive correlations with constructs related to the work environment. It has a version for university students and has been translated into several languages.

The Emotional Intelligence Questionnaire (EIQ) [68] is another tool designed to measure EI in the workplace. It has face, content, construct, and predictive validity, although the internal consistency of its scales varies between good and not very acceptable. Years later, the Emotional Intelligence Inventory [69] was developed in India. It was also designed to measure EI using a mixed concept in the workplace. It is made up of 10 dimensions, which have an internal consistency between acceptable and excellent. It has correlations with several related scales and with the number of promotions achieved and success in employment, which is proof of its predictive validity.

The Emotional Intelligence Appraisal (EIA) [70] is a set of surveys that measures EI in the workplace using the four main components of the Goleman model. Their items have been evaluated by experts. It has an internal consistency between very good and excellent. It has three versions: an online self-report, an online multi-rater report (which is combined with responses from co-workers), and another one that has anonymous ratings from several people to get an EI score for the whole team. The Emotional Intelligence Scale (EIS) [71] is another tool based on the Goleman model. It is composed of three dimensions and it has excellent internal consistency. The content of the items has been validated by expert evaluations.

The USM Emotional Quotient Inventory (USMEQ-i) [72] is a tool developed in Malaysia. It consists of a total of seven dimensions composed of 46 items. Seven of these items make up the “faking index items”, that measure the tendency of respondents to manifest social desirability and have a very good internal consistency (*α* = 0.83). The reliability of the total instrument yields excellent values.

The Indigenous Scale of Emotional Intelligence [73] is a Pakistani instrument developed in the Urdu language. The final items were selected from an initial set after passing through the judgment of four experts based on the fidelity to the construct: clarity, redundancy, reliability, and compression. It has excellent internal consistency. Additionally, it presents construct validity (as women obtain higher scores than men) and correlations with the EQ-i.

Years later, the Mobile Emotional Intelligence Test (MEIT) was developed [64]. It is a Spanish instrument used to measure EI online in work contexts. It is made up of seven tasks (perceptive tasks and identification tasks) to assess the emotional perception of both others and oneself, respectively, face task, in which the most appropriate photograph related to the demanded emotion must be chosen, three comprehension tasks (composition, deduction and retrospective), and story task, in which participants must choose the best action to manage feelings in a given story. It presents excellent internal consistency and convergent validity.

### 3.3. Trait-Based Measures

This category is composed of trait-based instruments. The Trait Emotional Intelligence Questionnaire (TEIQue) [6] is the main instrument of this model. It is a tool widely used in many countries. It has excellent internal consistency and it shows significant correlations with the Big Five Personality. It has a short version, a 360-degree version, a version for children and another one for teenagers. It has been translated into many languages.

Years later, the Rotterdam Emotional Intelligence Scale (REIS) [75] was developed, the other instrument belonging to this category. It is a self-report instrument designed in Dutch. It has a very good internal consistency and it presents correlations with WEIS, TEIQue and PEC and its validity criterion has also been tested.

### 3.4. Measures Based on Other Models

Some instruments cannot be included within these categories since they have been conceptualized under different models. The first one is the Genos Emotional Intelligence Inventory [76], previously known as SUEIT. It is based on an original model. It was specifically designed for use in the workplace, but it does not measure EI per se, but rather the frequency with which people display a variety of emotionally intelligent behaviours in the workplace. It presents very good reliability and convergent and predictive validity. In addition, it has two reduced versions.

The Profile of Emotional Competence (PEC) [77] is based on the model of Mikolajczak [83], which replicates the four dimensions proposed by Mayer and Salovey but separates the identification from the expression of the emotions and distinguishes the intrapersonal aspect from the interpersonal aspect of each dimension. It contains two main scales, and has excellent internal consistency and convergent, divergent and criterion validity. The original one was developed in French, but it has been translated into several languages.

The last of the instruments identified is the Group-level Emotional Intelligence Questionnaire [79]. It was designed in the USA to assess EI in work groups under Ghuman’s theoretical model [79]. This model conceives EI as a two-component construct: group relationship capability (GRC) and group emotional capability (GEC). All of them have very good internal consistency.

Regarding the framework of the Standards, differences were found among them, resulting in an unequal distribution throughout the articles. The percentages of each type of validity can be seen in Table 2.

The instruments whose original sources could not be retrieved are cited in Table 3. The main reasons were that they were articles from books to which the authors did not have access, unpublished documents or documents with restricted access.

## 4. Discussion

The main aim of this study is to offer an updated systematic review of EI instruments in order to provide researchers and professionals with a list of tools that can be applied in the professional field with their characteristics, psychometric properties and versions, as well as a brief description of the instrument. For this purpose, a systematic review of the scientific literature on EI has been carried out using the WoS database through a search of all articles published between 1900 and the present.

The number of instruments developed has been increasing in recent years. In the 1990s barely any instruments were developed and their production was limited to approximately one per year and to practically one country (i.e., the USA). This may be due to the recent conceptualisation of EI, as well as to the difficulty that researchers found in constructing emotion-centred questions with objective criteria [15]. However, over the years, the production of instruments to measure EI has been increasing and, in addition, it has been extended to other geographical areas. This may be due to the importance that EI has reached over the years in multiple areas (e.g., health, organizational, educational, etc.). With the passage of time, and the introduction of new technologies, multimedia platforms have begun to be used to present stimuli to participants. Recent research in EI has determined that emotions are expressed and perceived through visual and auditory signals (i.e., the tone of voice and the dynamic movements of the face and body) [94]. Thus, a meta-analysis revealed that video-based tests tend to have a higher criterion-related validity than text-based stimuli [95].

Regarding the results, a total of 40 instruments produced from 1995 to 2020 have been located. The instruments registered in a greater number of studies, and that have been most used over the years are EQ-i, SSRI, MSCEIT 2.0, TMMS, WLEIS, and TEIQue. These tools have the largest number of versions (e.g., reduced or for different ages or contexts) and are the ones that have been validated in more languages. The most recent instruments hardly have translations apart from their original version, and they have been tested on very few occasions. Most of the articles have not been developed for a specific context.

On the other hand, as can be seen in the results, most of the instruments are grouped under the three main conceptual models described in the introduction (ability, trait and mixed). These models are vertebrated around the construct of EI. However, they present differences in the way of conceptualizing it and, therefore, also of measuring it. For example, the ability-based concept of EI is measured by maximum performance tests while trait-based EI is measured by self-report questionnaires. This may, in itself, lead to different outcomes, even if the underlying model used is the same [96,97].

The ability model, introduced by Mayer and Salovey, is composed of other hierarchically ordered abilities, in which the understanding and management dimensions involve higher-order cognitive processes (strategic), and are based on perception and facilitation, which involve instantaneous processing of emotional information (experiential) [4]. This model has received wide recognition and has served as a basis for the development of other models. However, it has been questioned through factor analysis that does not support a hierarchical model with an underlying global EI factor. Furthermore, emotional thought facilitation (second dimension) did not arise as a separate factor and was found to be empirically redundant with the other branches [96].

Intelligence and personality researchers have questioned the very existence of ability EI, and they suggest that it is nothing more than intelligence. This fact is supported by the high correlations found between ability-based EI and the intellectual quotient [15,96]. On the other hand, there is the possibility of falsifying the results by responding strategically for the purpose of social desirability. However, one of the advantages of the ability model is that, through the maximum performance tests, it is not possible to adulterate them. This is because participants must choose the answer they think is correct to get the highest possible score. Another advantage is that these types of instruments tend to be more attractive because they are made up of tests in which it is required to resolve problems, solve puzzles, perform comprehension tasks or choose images [15].

The Petrides and Furnham model [5] emerged as an alternative to the ability-based model and is related to dispositional tendencies, personality traits, or self-efficacy beliefs that are measured by self-report tests. The tools based on this model are not exempt from criticism. These instruments present a number of disadvantages, the most frequently cited are being vulnerability to counterfeiting and social desirability [96]. The participant can obtain a high EI profile by responding in a strategically and socially desirable way, especially when they are examined in work contexts by supervisors or in job interviews. People are not always good judges of their emotional abilities [98], and may tend to unintentionally underestimate or overestimate their EI. Another criticism of self-report tools is their ecological validity (i.e., external validity that analyses the test environment and determines how much it influences the results) [96].

On the contrary, the fact that such tools do not present correct or incorrect answers can be advantageous in certain cases. High EI trait scores are not necessarily adaptive or low maladaptive. That is, self-report tools give rise to emotional profiles that simply fit better and are more advantageous in some contexts than in others [97]. On the other hand, trait-based tools have demonstrated good incremental validity over cognitive intelligence and personality compared to ability-based EI tests [99]. Furthermore, they tend to have very good psychometric properties, have no questionable theoretical basis, and are moderately and significantly correlate with a large set of outcome variables [15].

One aspect observed in this systematic review is that the main measure of the estimated reliability in the analysed studies has been internal consistency. However, this estimate is not interchangeable with other measurement error estimates. This coefficient gives a photographic picture of the measurement error and does not include variability over time. There are other reliability indicators (e.g., stability or test–retest) that are more relevant for social intervention purposes [100], and that according to the estimation design, can differentiate into trait variability or state variability, that is, respectively stability and dependability [101]. It has been found that the use of stability measures as a reliability parameter is not frequent. In methodological and substantive contexts, reproducibility is essential for the advancement of knowledge. For this reason, it is necessary to identify measures that can be used as parameters to compare the results of different studies [102]. On the other hand, the standard coefficient of internal consistency has been coefficient *α* [103]. This measure has been questioned in relation to its apparent misinformed use of its restrictions [104,105,106], of which Cronbach himself highlighted its limited applications [104]. Other reliability measures have been recommended (e.g., ω) [107], and the reliability estimation practice in the creation of EI measurements needs to be updated. Usually, ω estimation is integrated into the modelling-based estimation, where SEM or IRT methodology is required to corroborate the internal structure of the score [108,109,110] and extract the parameters used to calculate reliability (i.e., factorial loads).

Another methodological aspect to highlight is that predominantly, the construction of EI measures was based on linear modelling or classical test theory. In contrast, the least used approach was item response theory (IRT), which provides other descriptive and evaluative parameters of the quality of the score measurement, such as the information function or the characteristic curves of the options, among others.

On the other hand, it is striking that some of the articles found prove the construct validity of their instruments by obtaining higher EI scores by women than men [56,58,73]. This has also been seen in the scientific literature and in research such as that of Fischer et al. [111], in which it was found that women tend to score higher in EI tests or empathy tests than men, especially, but not only, if it is measured through self-report. Additionally, striking is the study by Molero et al. [112], in which significant differences were observed among the various EI components between men and women. However, this is not the case in all the articles analysed in this study, nor in all the most current scientific literature. This fact has led to the development of different hypotheses about how far, why, and under what circumstances women could outperform men. There are several theories that have emerged around it. There is one that claims that these differences could be related to different modes of emotional processing in the brain [113,114]. Another theory points to possible differences in emotional perception that suggest that women are more accurate than men in this process when facial manifestations of emotion are subtle, but not when stimuli are highly expressive [115]. Additionally, another one points out that the expression of emotions is consistent with sex, which may be influenced by contextual factors, including the immediate social context and broader cultural contexts [116]. However, other variables such as age or years of experience in the position should also be taken into account. For example, the study by Miguel-Torres et al. [117] showed a better ability to feel, express, and understand emotional states in younger nurses, while the ability to regulate emotions was greater in those who had worked for more years. For this reason, nowadays firm conclusions cannot be drawn and it must be taken into account that the differences found are generally small. Thus, more research is needed on the differences that may exist between men and women in the processes of perception, expression and emotional management before establishing possible social implications of these findings.

### 4.1. Limitations

This study is not without limitations. Some are inherent in this type of studies, such as publication bias (i.e., the non-publication of studies with results that do not show significant differences) that could have resulted in a loss of articles that have not been published and that used instruments other than those found. In addition, instruments that could not be accessed from their original manuscript could not be included in the systematic review. On the other hand, despite the advantages of WoS, the fact that the search was conducted in a single database may lead to some loss of literature. Furthermore, the systematic review was restricted to peer-reviewed publications and thus different studies may be presented in other information sources, such as books or grey literature. Articles that were in the press and those that may have been published in the course of the compilation of this study have not been collected either. Additionally, the entire process of searching for references was carried out by only one investigator, so an estimate of inter-judge reliability cannot be made, as well as data extraction. There are many aspects of the PRISMA statement that, due to the purpose of our research, our study does not include (visible as NA in Table A1). However, it is necessary to develop a protocol for recording the inclusion and exclusion criteria of the primary studies to prevent bias (e.g., bias in the selection process). There are also some methodological aspects to be improved, such as the lack of methods used to assess the risk of bias in the included studies, the preparation or synthesis of the data, or the certainty in the body of evidence of a result. In future research it is necessary to take into account and develop these aspects in order to improve the replicability and methodological validity of the study, and to facilitate the transparency of the research process. In contrast to the above, one of the strengths of this study was to minimize the presence of biases that could alter the results. To minimize language bias, articles submitted in any language were searched for and accepted to avoid over-presentation of studies in one language, and under-presentation in others [20]. In addition, this study takes into account and exposes five sources of evidence of validity of the instruments through the Standards: content, response processes, internal structure, relationship with other variables and the consequences of testing. Other aspects to be improved in the future include performing the same search in other databases such as EBSCO and Scopus to obtain possible articles not covered in WoS. A manual search for additional articles would also be useful, for example, in the references of other articles or in the grey literature.

### 4.2. Practical Implication

The relationship between EI and personal development has been of great interest in psychological research over time [8]. A good study of the instruments that measure constructs such as EI can be of great help both in the field of prevention and psychological intervention in social settings. The revision of EI instruments is intended to contribute to facilitating work in the general population in a way that the development of EI is promoted and antisocial behaviours are reduced. In addition, since it correlates with variables that serve as protectors against psychological distress, this work also contributes to improving, in some cases, the general level of health.

Through this systematic review, we can see the great effort that has been made by researchers not only to improve existing EI measurement instruments, but also in the construction of new instruments that help professionals in the educational, business and health fields, as well as the general population. However, given the rapid changes that society is experiencing, partly due to the effects of modernization and technology, there is a demand to go beyond measurement. For example, from educational and business institutions and from family and community organizations it is necessary to promote activities, support and commitment towards actions oriented to EI under the consideration that this construct can be improved at any age and that it increases with experience.

## 5. Conclusions

From the results obtained in this study, numerous instruments have been found that can be used to measure EI in professionals. Over the years, the production of instruments to measure EI has been increasing and, moreover, has spread to other geographical areas. The most recent instruments have hardly been translated beyond their original version and have been tested very rarely. In order for future research to benefit from these new instruments, a greater number of uses in larger samples and in other contexts would be desirable.

In addition, most of the instruments are grouped under the three main conceptual models described in the introduction (ability, trait and mixed). Each model has a number of advantages and disadvantages. In the ability model it is not possible to adulterate the results by strategic responses and they tend to be more attractive tests; however, factor analyses do not support a hierarchical model with an underlying global EI factor. The trait-based model, on the other hand, employs measures that have no right or wrong answers, so they result in emotional profiles that are more advantageous in some contexts than others, and they tend to have very good psychometric properties. However, they are susceptible to falsification and social desirability.

On the other hand, it is necessary to identify measures that can be used as parameters to compare the results of different studies. In addition, the standard coefficient of internal consistency has been the α coefficient, which has been questioned in relation to its apparent misinformed use of its restrictions. It would be advisable to use other reliability measures and to update the reliability estimation practice in the creation of EI measures.

Finally, some of the articles found test the construct validity of their instruments by obtaining higher EI scores from women than from men. Different hypotheses have been developed about to what extent, why and under what circumstances women would outperform men; differences may be related to different modes of emotional processing in the brain or possible differences in emotional perception or to the influence of contextual factors. However, it would be interesting to further investigate the differences that may exist between men and women or to take into account other factors such as age or number of years of experience before establishing possible practical implications.

## Figures and Tables

**Figure 1 healthcare-09-01696-f001:**
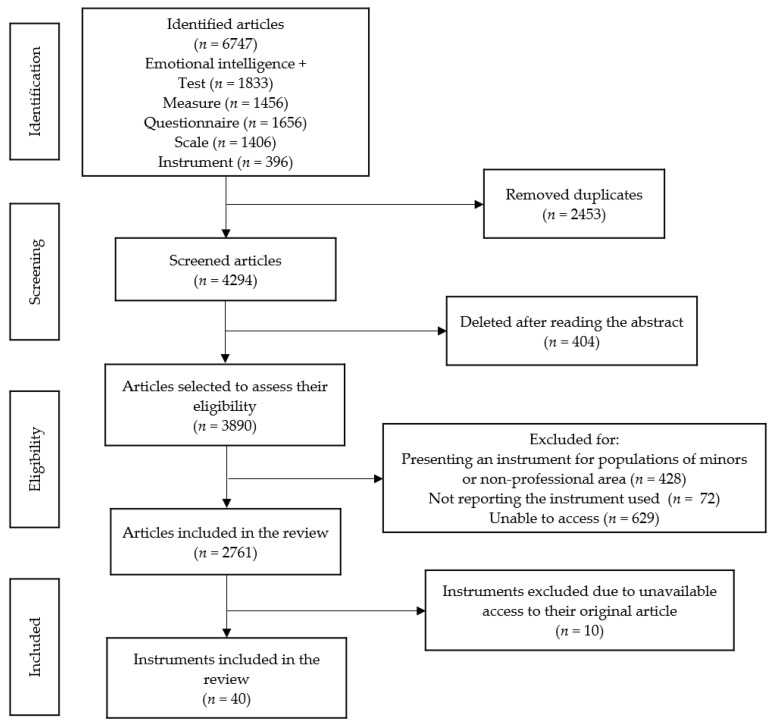
Flowchart according to PRISMA.

**Table 1 healthcare-09-01696-t001:** Main characteristics of the included instruments.

Measure	Structural Characteristics	Languages (Origin Country)	Psychometric Data		Other Versions	Last Validation
			Reliability	Validity		
Ability-Based Model						
Trait Meta-Mood Scale (TMMS) [30]	Format: scale (five-point Likert) Num. items: 48 Dimensions and items: ·Attention to feelings (21) ·Emotional clarity (15) ·Repair of the emotions (12)	English (USA)	Internal consistency: *α* = 0.82–0.88 Test–retest: None	Convergent: (+): Self-Consciousness Scale, optimism (LOT) and beliefs about the changeability of negative moods (CES-D), and the Expectancies for Negative Mood Regulation (−): ambivalence over emotional expression, depression	TMMS-30 version (recommended by the authors) TMMS-24 version (widely and internationally adapted and used) [31] Translated into several languages	Team-Trait Meta Mood Scale (T-TMMS) [32]
Schutte Self-Report Inventory (SSRI) [33]	Format: questionnaire (five-point Likert) Num. items: 33 Dimensions and items: **·**Appraisal and expression of emotion (13) **·**Regulation of emotion (10) **·**Utilization of emotion (10)	English (USA)	Internal consistency: *α* = 0.90 Test–retest: *r* = 0.78 (after 2 weeks)	Internal structure: Principal-components analysis Convergent: (+): attention to feelings and mood repair (TMMS), optimism (LOT), and openness to experience (BFP) (−): pessimism (LOT), TAS, ZDS, and BIS Predictive: Therapist scored significantly higher than prisoners, and scores significantly predicted grade point average at the end of the year of college students	Modified version by Austin et al. [34] Brief version-10 items by Davies et al. [35] Translated into several languages	Validation for pre-service physical education teachers [36]
Multifactor Emotional Intelligence Scale (MEIS) [37]	Format: scale Num. items: 402 Dimensions and items: **·**Perceiving emotion (186) **·**Assimilating emotion (88) **·**Understanding emotion (80) **·**Managing emotion (48)	English (USA)	Internal consistency: *α* = 0.49–0.94 Test–retest: None	Internal structure: Exploratory factor analysis Content: Scoring evaluated by consensus, experts, and target	Translated into several languages	–
Mayer–Salovey–Caruso Emotional Intelligence Test (MSCEIT) [38]	Format: test (five-point Likert and multiple-choice items with correct or incorrect answers) Num. items: 141 Dimensions and items: **·**Perceiving and identifying emotions **·**Facilitation of thought **·**Understanding emotions **·**Managing emotions	English (USA)	Internal consistency: *α* = 0.76–0.91 for the four branch scores for both methods Split-half = 0.93 and 0.91 for consensus and expert scoring, respectively Test–retest: *r* = 0.55–0.88 (after 3 weeks)	Content: The scoring is evaluated by consensus, and experts	MSCEIT Revised Version (MSCEIT 2.0) MSCEIT Youth Version (MSCEIT-YV)Translated into several languages	Traditional Chinese version (MSCEIT-TC) for people with schizophrenia [39]
Profile of Emotional Intelligence (PIEMO) [40]	Format: inventory (true and false answer options) Num. items: 161 Dimensions and items: **·**Impulse inhibition (25) **·**Empathy (17) **·**Optimism (28) **·**Social skills (16) **·**Emotional expression (14) **·**Achievement’s acknowledgement (23) **·**Self-esteem (27) **·**Kindness (11)	Spanish (Mexico)	Internal consistency: *α* = 0.96 Test–retest: None	Internal structure: Confirmatory factor analysis Content: Experts asked about the items	–	–
Wong and Law’s Emotional Intelligence Scale (WLEIS) [41]	Format: scale (7-point Likert) Num. items: 16 Dimensions and items: **·**Self-emotional appraisal (4) **·**Others’ emotional appraisal (4) **·**Regulation of emotion (4) **·**Use of emotion (4)	English (China)	Internal consistency: *α* = 0.76–0.89 Test–retest: None	Internal structure: Exploratory factor analysis Convergent: (+): EQ-i Discriminant: Not correlated with BFP	Translated into several languages	Korean version for Nurses [42]
Workgroup Emotional Intelligence Profile-3 (WEIP-3) [43]	Format: scale (7-point Likert) Num. items: 27 Dimensions and items: **·**Awareness of own emotions **·**Ability to discuss own emotions **·**Ability to use own emotions to facilitate thinking **·**Ability to recognise others’ emotions **·**Ability to detect false displays of emotion in others **·**Empathetic concern **·**Ability to manage others’ emotions	English (Australia)	Internal consistency: *α* = 0.86 Test–retest: None	Internal structure: Exploratory factor analysis Convergent: (+): Revised Self-Monitoring Scale, TMMS, IRI, and JABRI	Workgroup Emotional Intelligence Profile-Short version (WEIP-S) Later versions Translated into few languages	Spanish version of the short version (WEIP-S) in the sports context [44]
Multidimensional Emotional Intelligence Assessment (MEIA) [45]	Format: scale (6-point Likert) Num. items: 150 Dimensions and items: **·**Recognition of emotion in the self **·**Nonverbal emotional expression **·**Recognition of emotion in others **·**Empathy **·**Regulation of emotion in the self **·**Regulation of emotion in others **·**Intuition versus reason **·**Creative thinking **·**Mood redirected attention **·**Motivating emotions	English (USA)	Internal consistency: *α* = 0.81 Test–retest: *r* = 0.67–0.88 (after 4–6 weeks)	Internal structure: Principal component analysis Convergent: (+/−): JPI-R Content: Retained only items judged a priori as representing a particular construct Criterion: (+): three satisfaction measures are consistent with the corresponding reported results for other self-report EI scales	Multidimensional Emotional Intelligence Assessment —Workplace (MEIA-W)	Multidimensional Emotional Intelligence Assessment—Workplace—Revised (MEIA-W-R; 2006, unpublished)
Emotional Intelligence Questionnaire (EmIn) [46]	Format: scale (4-point Likert) Num. items: 40 Dimensions and items: **·**Interpersonal EI **·**Intrapersonal EI	Russian (Russia)	Internal consistency: *α* = 0.76–0.78 Test–retest: None	Internal structure: Factor analysis	–	–
Sojo and Steinkopf Emotional Intelligence Inventory—Revised version (IIESS-R) [47]	Format: inventory Num. items: 34 Dimensions and items: **·**Perception of emotions in other people (11) **·**Perception of own emotions (11) **·**Emotion management (12)	Spanish (Venezuela)	Internal consistency: *α* = 0.90 Test–retest: None	Internal structure: Exploratory factor analysis Principal component analysis Convergent: (+/−): IRI, and Scale of Emotional Sensitivity Content: Content of items reviewed by expert judges	–	–
Self-Rated Emotional Intelligence Scale (SREIS) [48]	Format: scale (five-point Likert) Num. items: 19 Dimensions and items: **·**Perceiving emotions (4) **·**Using emotions (3) **·**Understanding emotions (4) **·**Managing emotions (8)	English (USA)	Internal consistency: *α* = 0.84 Test–retest: None	Internal structure: Confirmatory factor analysis Content: Before the administration, graduate students familiar with Mayer and Salovey’s (1997) model of EI rated the validity of each item	–	–
Emotional Intelligence Self-Description Inventory (EISDI) [49]	Format: inventory (7-point Likert) Num. items: 24 Dimensions and items: **·**Perception and appraisal of emotions (6) **·**Facilitating thinking with emotions (6) **·**Understanding emotion (6) **·**Regulation and management of emotion (6)	English (USA)	Internal consistency: *α* = 0.91 Test–retest: *r* = 0.75–0.83 (after 2 weeks)	Internal structure: Confirmatory factor analysis Convergent: (+): WLEIS and SREIS (+/−): BFP Discriminant: Acceptable discriminant validity vis-à-vis the Big Five Personality variables because of the criticism from scholars that EI is “little more than a repackaging of personality characteristics”	–	–
Greek Emotional Intelligence Scale (GEIS) [50]	Format: scale Num. items: 52 Dimensions and items: **·**Expression and recognition of emotions (15) **·**Control of emotions (15) **·**Use of emotions for facilitating thinking (12) **·**Caring and empathy (10)	Greek (Greece)	Internal consistency: *α* = 0.89 Test–retest: *r* = 0.90 (after 2 weeks)	Internal structure: Principal component analysis Convergent: (+/−): BFP, SSRI, TAS, TMMS, SSI, EES, SWLS, PANAS, Locus of Control, and ASSET	–	–
Situational Test of Emotion Management (STEM) [51]	Format: test (multiple-choice/rate the extent) Num. items: 44 items Dimensions and items: **·**Anger (18) **·**Sadness (14) **·**Fear (12)	English (Australia)	Internal consistency: *α* = 0.68 (multiple choice) *α* = 0.92 (rate the extent) Test–retest: None	Convergent: (+): multiple-choice STEM with Vocabulary test, agreeableness (OCEANIC-20), and retrospective (SWLS) (−): externally oriented thinking (TAS-20) Criterion: (+): multiple-choice STEM with psychology grade, and weighted average mark	Situational Test of Emotional Management-brief version Translated into few languages	STEM-B in Chinese context [52]
Situational Test of Emotional Understanding (STEU) [51]	Format: test (multiple-choice items) Num. items: 42 Dimensions and items: **·**Context-reduced (14) **·**Personal-life context (14) **·**Workplace context (14)	English (Australia)	Internal consistency: *α* = 0.71 Test–retest: None	Convergent: (+): STEM (multiple choice and rate the extent; Stories (MEIS), Vocabulary test, and agreeableness (OCEANIC-20) (−): externally oriented thinking (TAS-20) Criterion: (+): psychology grade, and weighted average mark	Situational Test of Emotional Understanding-brief version Translated into few languages	STEU-B in Chinese context [52]
Emotional Skills and Competence Questionnaire (ESCQ) [53]	Format: questionnaire (five-point Likert) Num. items: 45 Dimensions and items: **·**Perceive and understand emotions (15) **·**Express and label emotions (14) **·**Manage and regulate emotions (16)	Croatian (Croatia)	Internal consistency: *α* = 0.67–0.90 Test–retest: None	Internal Structure: Confirmatory factor analysis Convergent: (+): SSRI, SSI, and BFP (−): TAS	Translated into several languages	Portuguese academic context [54]
Audiovisual Test of Emotional Intelligence (AVEI) [55]	Format: test (multiple-choice items with correct or incorrect answers) Num. items: 27 Dimensions and items: **·**Love **·**Pride **·**Shame **·**Anger **·**Frustration **·**Happiness **·**Care **·**Fear **·**Satisfaction **·**Anger **·**Sadness **·**Envy	English (Israel)	Intraclass correlation: ICC = 0.65 Test–retest: None	Content: Experts asked about the items Criterion: (+): academic achievement, psychometric exam score, clinical practice grade, and interpersonal skill workshop grade (measures that are traditionally considered to be proxies of cognitive mental abilities)	–	–
Geneva Emotion Recognition Test (GERT) [56]	Format: test (forced-choice format) Num. items: 83 Dimensions and items: **·**Amusement (6) **·**Irritation (6) **·**Anger (6) **·**Joy (6) **·**Disgust (6) **·**Fear (6) **·**Despair (5) **·**Pleasure (6) **·**Pride (6) **·**Relief (6) **·**Anxiety (6) **·**Surprise (6) **·**Interest (6) **·**Sadness (6)	German (Germany)	IRT parameters (*ρ* = 0.92) Test–retest: None	Internal structure: Comparative factor analysis Ecological: Multimodal stimuli; videos portrayed by 10 actors, men and women, and of different ages Construct: Women scored significantly higher than men	Geneva Emotion Recognition Test short version (GERT-S) Translated into few languages	Geneva Emotional Competence Test (GECo) workplace context [57]
Test of Emotional Intelligence (TIE) [58]	Format: test (five-point Likert) Num. items: 24 Dimensions and items: **·**Perception (6) **·**Understanding (6) **·**Facilitation (6) **·**Management (6)	Polish (Poland)	Internal consistency: *α* = 0.88 Test–retest: None	Convergent: (+): SSRI and SIE-T Discriminant: Not correlated with NEO-FFI Construct: Women scored significantly higher than men	–	–
Videotest of Emotion Recognition [59]	Format: test (6-point Likert) Num. items: 15 Dimensions and items: **·**Anger (1) **·**Displeasure (1) **·**Relaxation (1) **·**Arousal (1) **·**Surprise (1) **·**Suffering (1) **·**Contempt (1) **·**Happiness (1) **·**Shame (1) **·**Fear (1) **·**Anxiety (1) **·**Calmness (1) **·**Disgust (1) **·**Guilt (1) **·**Interest (1)	Russian (Russia)	Internal consistency: *α* = 0.74 Test–retest: *r* = 0.55	Convergent: (+): MSCEIT and EmIn	–	–
Self-Perception of Emotional Intelligence Questionnaire (EIQ-SP) [60]	Format: questionnaire (five-point Likert) Num. items: 18 Dimensions and items: **·**Perception, evaluation and emotional expression (4) **·**Emotional facilitation of thought (5) **·**Emotional understanding and analysis (6) **·**Emotion regulation (3)	Portuguese (Portugal)	Internal consistency: *α* = 0.70–0.77 Test–retest: None	Internal structure: Exploratory factor analysis Confirmatory factor analysis	–	–
Three-Branch Emotional Intelligence Forced-Choice Assessment (TEIFA) [61]	Format: forced-choice assessment Num. items: 18 Dimensions and items: **·**Emotion perception (6) **·**Emotion understanding (6) **·**Emotion management (6)	English (USA)	Reliability of TEIFA is not reported as reliability for forced-choice tests is artificially high	Internal structure: Confirmatory factor analysis Convergent: (+/−): SSRI	–	–
Three-Branch Emotional Intelligence Rating Scale Assessment (TEIRA) [61]	Format: scale (6-point Likert) Num. items: 18 Dimensions and items: **·**Emotion perception (6) **·**Emotion understanding (6) **·**Emotion management (6)	English (USA)	Internal consistency: *α* = 0.79–0.90 Test–retest: None	Internal structure: Confirmatory factor analysis Convergent: (+): STEU-B, STEM-B and SREIS	–	–
North Dakota Emotional Abilities Test (NEAT) [62]	Format: test (rate-the-extent) Num. items: 30 Dimensions and items: **·**Perception (10) **·**Understanding (10) **·**Management (10)	English (USA)	Internal consistency: *α* = 0.74–0.90 Test–retest: None	Internal structure: Confirmatory factor analysis Predictive: NEAT scores predicted the ability to decode facial expressions of emotion, the ability to assign accurate evaluations to word stimuli, and the ability to make judgments consistent with appraisal theories of emotion Convergent: (+): DANVA 2-AF, STEU and STEM	–	–
Perceived Emotional Intelligence Inventory (IIEP) [63]	Format: inventory (five-point Likert) Num. items: 101 Dimensions and items: **·**Emotional attention (interpersonal) (21) **·**Emotional understanding (intrapersonal) (20) **·**Emotional regulation (intrapersonal) (22) **·**Emotional attention (intrapersonal) (13) **·**Emotional understanding and regulation (interpersonal) (13) **·**Emotional expression (12)	Spanish (Argentina)	Internal consistency: *α* = 0.81–0.93 Test–retest: None	Internal structure: Exploratory factor analysis Content: Judges asked to classify each item according to the dimensions evaluated, judge each item considering its relevance and formal quality, and make all necessary observations and suggestions in order to improve them	–	–
Mobile Emotional Intelligence Test (MEIT) [64]	Format: test (different tasks) Num. items: 42 Dimensions and items: **·**Perceiving emotions **·**Understanding emotions **·**Managing emotions	Spanish (Spain)	Internal consistency: *α* = 0.91 Test–retest: None	Internal structure: Confirmatory factor analysis Convergent: (+): TMMS-24, RAVEN and SWLS	–	–
Emotional Intelligence Test (EIT) [65]	Format: test Num. items: Dimensions and items: **·**Perceiving emotions **·**Facilitation of thought using emotions **·**Understating and analyzing emotions **·**Conscious managing of emotions	Russian (Russia)	Internal consistency: *α* = 0.93 Test–retest: None	Internal structure: Factor analysis Convergent: (+): MSCEIT 2.0	–	–
**Mixed Model**						
Emotional Quotient Inventory (EQ-i) [7]	Format: inventory (five-point Likert) Num. items: 133 Dimensions and items: **·**Intrapersonal **·**Interpersonal **·**Adaptability **·**Stress management **·**General mood	English (USA)	Internal consistency: *α* = 0.75–0.84 Test–retest: None	Internal structure: Principal component analysis Construct: (+): measures of emotional stability (−): measures of neuroticism and psychopathology	EQ-i: Short Version (EQ-i: S) EQ-i 2.0 EQ-i: 360° Version (EQ-i: 360°) EQ-i: Youth Version (EQ-i: YV) and EQ-i: Youth Short Version (EQ-i: YVS) Translated into more than 30 languages	EQ-i: YV in Spanish adolescents with Down syndrome [66]
Emotional Competence Inventory 2.0, (ECI 2.0, previously ECI) [67]	Format: inventory (6-point Likert) Num. items: 72 Dimensions and items: **·**Self-awareness (18) **·**Self-management (18) **·**Social awareness (18) **·**Relationship management (18)	English (USA)	Internal consistency for “others” ratings: *α* = 0.78 Internal consistency for “self” ratings: *α* = 0.63 Test–retest: None	Internal structure: Confirmatory factor analysis	ECI (older version) ECI-University Version (ECI-U)	–
Emotional Intelligence Questionnaire (EIQ) [68]	Format: questionnaire Num. items: 69 Dimensions and items: **·**Self-awareness (12) **·**Emotional resilience (11) **·**Motivation (10) **·**Interpersonal sensitivity (12) **·**Influence (10) **·**Decisiveness (7) **·**Conscientiousness and integrity (7)	English (UK)	Internal consistency: *α* = 0.70–0.59 Split-half = 0.52–0.71 Test–retest: None	Face: Adverse comments not received and many subjects said that the questionnaire was measuring EI Content: Extensive literature revised about aspects of EI Construct: (+/−): 16PF, OPQ, and BTR Predictive: EQ competences scale predicted organisational level advancement over a seven-year period	–	–
Emotional Intelligence Inventory [69]	Format: inventory (7-point Likert) Num. items: 61 Dimensions and items: **·**Emotionality and impulsiveness (15) **·**Self-acceptance (5) **·**Problem-solving orientation (6) **·**Self- awareness (6) **·**Self-confidence (4) **·**Decisiveness and independence (7) **·**Personal fulfilment (4) **·**Empathy (4) **·**Anxiety and stress (7) **·**Assertiveness (3)	English (India)	Internal consistency: *α* = 0.76–0.78 Test–retest: None	Predictive: (+): several scales and number of promotions attained and rated job success	–	–
Emotional Intelligence Appraisal (EIA) [70]	Format: test (6-point Likert) Num. items: 28 Dimensions and items: **·**Self-awareness (6) **·**Social awareness (5) **·**Self-management (9) **·**Relationship management (8)	English (USA)	Internal consistency: *α* = 0.85–0.91 Test–retest: None	Internal structure: Principal component analysis Content: Experts asked about the items	Me Edition (online self-report version) MR Edition (online multi-rater method with combination of responses from co-workers) Team EQ Edition (anonymous ratings from multiple individuals to yield an EQ score for the entire team)	–
Emotional Intelligence Scale (EIS) [71]	Format: scale (4-point Likert) Num. items: 23 Dimensions and items: **·**Self-management and creativity **·**Social capacity **·**Emotional self-awareness	English (Norway)	Internal consistency: *α* = 0.93 Test–retest: None	Internal structure: Exploratory factor analysis Content: Tested by means of expert evaluation	–	–
USM Emotional Quotient Inventory (USMEQ-i) [72]	Format: inventory (five-point Likert) Num. items: 46 Dimensions and items: **·**Emotional control **·**Emotional maturity **·**Emotional conscientiousness **·**Emotional awareness **·**Emotional commitment **·**Emotional fortitude **·**Emotional expression	Malaysian (Malaysia)	Internal consistency: *α* = 0.96 Test–retest: None	Internal structure: Factor analysis	–	–
Indigenous Scale of Emotional Intelligence [73]	Format: scale (4-point Likert) Num. items: 56 Dimensions and items: **·**Interpersonal skill (8) **·**Self-regard (6) **·**Assertiveness (7) **·**Emotional self-awareness (5) **·**Empathy (5) **·**Impulse control (5) **·**Flexibility (5) **·**Problem solving (5) **·**Stress tolerance (5) **·**Optimism (5)	Urdu (Pakistan)	Internal consistency: *α* = 0.95 Test–retest: None	Internal structure: Principal component analysis Construct: Women scored significantly higher than men Convergent: (+): EQ-i	–	–
**Trait-Based Model**						
Trait Emotional Intelligence Questionnaire (TEIQue) [6]	Format: questionnaire (five-point Likert) Num. items: 153 Dimensions and items: **·**Emotionality **·**Self-control **·**Sociality **·**Well-being	English (UK)	Internal consistency: *α* = 0.89–0.92 Test–retest: None	Internal structure: Principal component analysis Convergent: (+): BFP	TEIQue Short Form (TEIQue-SF) TEIQue-360° and 360°-SF TEIQue Adolescent Form (TEIQue-AF) and TEIQue-ASF TEIQue Child Form (TEIQue-CF) Translated into several languages	Spanish-Chilean short form [74]
Rotterdam Emotional Intelligence Scale (REIS) [75]	Format: scale (five-point Likert) Num. items: 28 Dimensions and items: **·**Self-focused emotion appraisal (7) **·**Other-focused emotion appraisal (7) **·**Self-focused emotion regulation (7) **·**Other-focused emotion regulation (7)	Dutch (Netherlands)	Internal consistency: *α* = 0.80–0.85 Test–retest: None	Internal structure: Confirmatory factor analysis Convergent: (+): WEIS, TEIQue, and PEC Criterion: (−): self-focused emotion regulation with tutors’ perceived stress (+): other-focused emotion regulation with tutors’ work engagement, jobseekers’ other-rated interview performance and leaders’ transformational leadership style	–	–
**Others**						
Genos Emotional Intelligence Inventory (previously SUIET) [76]	Format: inventory (five-point Likert) Num. items: 70 Dimensions and items: **·**Emotional self-awareness (10) **·**Emotional expression (10) **·**Emotional awareness of others (10) **·**Emotional reasoning (10) **·**Emotional self-management (10) **·**Emotional management of others (10) **·**Emotional self-control (10)	English (Australia)	Internal consistency: *α* = 0.96 Test–retest: *r* = 0.83 (after 2 month) *r* = 0.72 (after 6 month)	Internal Structure: Confirmatory factor analysis Convergent: (+): SUEIT and TMMS Predictive: (+): performance (i.e., sales revenue) in a sample of pharmaceutical sales representatives	31-item Concise Version 14-item Short Version	–
Profile of Emotional Competence (PEC) [77]	Format: scale (five-point Likert) Num. items: 50 Dimensions and items: **·**Intrapersonal emotional competence (25) **·**Interpersonal emotional competence (25)	French (France)	Internal consistency: *α* = 0.93 Test–retest: None	Convergent: (+): TEIQue-SF Criterion: (+): happiness, subjective health, social relationships, and positive affectivity (−): negative affectivity Divergent: Not correlated with general cognitive ability	Translated into few languages	French short version for cancer patients [78]
Group-level Emotional Intelligence Questionnaire [79]	Format: questionnaire (five-point Likert) Num. items: 36 Dimensions and items: **·**Group learning ability (11) **·**Emotional capability (9) **·**Performance (5) **·**Relationship capability (9) **·**New member conformity (2)	English (USA)	Internal consistency: *α* = 0.80 Test–retest: None	Internal structure: Exploratory factor analysis Confirmatory factor analysis	–	–

TMMS: Trait Meta-Mood Scale, LOT: Life Orientation Test, CES-D: Center for Epidemiologic Studies Depression Scale; SSRI: Schutte Self-Report Inventory, BFP: Big Five Personality, TAS: Toronto Alexithymia Scale, ZDS: Zung Self-Rating Depression Scale, BIS: Barratt Impulsiveness Scale; MEIS: Multifactor Emotional Intelligence Scale; MSCEIT: Mayer-Salovey-Caruso Emotional Intelligence Test, MSCEIT 2.0: Mayer-Salovey-Caruso Emotional Intelligence Revised Version, MSCEIT-YV: Mayer-Salovey-Caruso Emotional Intelligence Youth Version, MSCEIT-TC: Mayer-Salovey-Caruso Emotional Intelligence Chinese Version; PIEMO: Profile of Emotional Intelligence; WLEIS: Wong and Law’s Emotional Intelligence Scale, EQ-i: Emotional Quotient Inventory; WEIP-3: Workgroup Emotional Intelligence Profile-3, WEIP-S: Workgroup Emotional Intelligence Profile-Short Version, IRI: Interpersonal Reactivity Index, JABRI: Job Associate-Bisociate Review Index; MEIA: Multidimensional Emotional Intelligence Assessment, JPI-R: Jackson Personality Inventory-Revised, MEIA-W: Multidimensional Emotional Intelligence Assessment-Workplace, MEIA-W-R: Multidimensional Emotional Intelligence Assessment-Workplace-Revised; EmIn: Emotional Intelligence Questionnaire; IIESS-R: Sojo and Steinkopf Emotional Intelligence Inventory-Revised Version; SREIS: Self-Rated Emotional Intelligence Scale; EISDI: Emotional Intelligence Self-Description Inventory; GEIS: Greek Emotional Intelligence Scale, SSI: Social Skills Inventory, EES: Emotion Empathy Scale, SWLS: Satisfaction with Life Scale, PANAS: Positive and Negative Affect Schedule, ASSET: An Organisational Stress Screening Tool; STEM: Situational Test of Emotion Management; OCEANIC-20: Openness Conscientiousness Extraversion Agreeableness Neuroticism Index Condensed 20-item version, STEM-B: Situational Test of Emotion Management-Brief Version; STEU: Situational Test of Emotional Understanding, STEU-B: Situational Test of Emotional Understanding-Brief Version; ESCQ: Emotional Skills and Competence Questionnaire; AVEI: Audiovisual Test of Emotional Intelligence; GERT: Geneva Emotion Recognition Test, GERT-S: Geneva Emotion Recognition Test-Short Version, GECo: Geneva Emotional Competence Test; TIE: Test of Emotional Intelligence, SIE-T: Emotional Intelligence Scale-Faces, NEO-FFI: NEO Five-Factor Inventory; EIQ-SP: Self-Perception of Emotional Intelligence Questionnaire; TEIFA: Three-Branch Emotional Intelligence Forced-Choice Assessment; TEIRA: Three-Brach Emotional Intelligence Rating Scale Assessment; NEAT: North Dakota Emotional Abilities Test, DANVA 2-AF: Diagnostic Analysis of Nonverbal Accuracy-Adult Faces; IIEP: Perceived Emotional Intelligence Inventory; MEIT: Mobile Emotional Intelligence Test; RAVEN: Raven’s Progressive Matrices; EIT: Emotional Intelligence Test; EQ-i: S: Emotional Quotient Inventory Short Version, EQ-i: 2.0: Emotional Quotient Inventory Revised Version, EQ-i: 360°: Emotional Quotient Inventory-360-degree version; EQ-i: YV: Emotional Quotient Inventory-Youth Version, EQ-i: YVS: Emotional Quotient Inventory Youth Short Version; ECI 2.0: Emotional Competence Inventory 2.0, ECI-U: Emotional Competence Inventory University Version; EIQ: Emotional Intelligence Questionnaire; 16PF: Sixteen Personality Factor Questionnaire, OPQ: Occupational Personality Questionnaire, BTR: Belbin Team Roles; EIA: Emotional Intelligence Appraisal; EIS: Emotional Intelligence Scale; USMEQ-I: USM Emotional Quotient Inventory; TEIQue: Trait Emotional Intelligence Questionnaire, TEIQue-SF: Trait Emotional Intelligence Questionnaire-Short Form, TEIQue-360°: Trait Emotional Intelligence Questionnaire-360-degree version, TEIQue-AF: Trait Emotional Intelligence Questionnaire Adolescent Form, TEIQue-CF: Trait Emotional Intelligence Questionnaire-Child Form; REIS: Rotterdam Emotional Intelligence Scale, PEC: Profile of Emotional Competence.

**Table 2 healthcare-09-01696-t002:** Number of studies and percentages for each validity test.

Study	Content	Response Processes	Internal Structure				Relationship with Other Variables	Consequences of Testing
			Factorial Analysis	Reliability	Test– Retest	Invariance		
Yes	11 (27.5%)	1 (2.5%)	23 (57.5%)	40 (100%)	7 (17.5%)	17 (42.5%)	22 (55%)	5 (12.5%)
No	29 (72.5%)	39 (97.5%)	17 (42.5%)	0	33 (82.5%)	23 (57.5%)	18 (45%)	35 (87.5%)

**Table 3 healthcare-09-01696-t003:** Information of the non-accessible instruments.

Measure	Type of Source	Information Source	Model	Dimensions and Items
Emotional Intelligence Questionnaire(UEK-45) [84]	Book	Mitić, P., Nedeljković, J., Takšić, V., Sporiš, G., Stojiljković, N., & Milčić, L. (2020). Sports performance as a moderator of the relationship between coping strategy and emotional intelligence. Kinesiology, 52(2), 281–289. https://doi.org/10.26582/k.52.2.15 (accessed on 7 July 2021)	Unknown	Dimensions: 3 Items: 45
Emotional Intelligence Questionnaire [85]	Book	Daryani, S., Aali, S., Amini, A., & Shareghi, B. (2017). A comparative study of the impact of emotional, cultural, and ethical intelligence of managers on improving bank performance. International Journal of Organizational Leadership, 6, 197–210. https://ijol.cikd.ca/article_60318_131fe99b0de8ccb1e59ec16f60d760f9.pdf (accessed on 7 July 2021)	Mixed	Dimensions: 6 Items: unknown
EQ Self-Assessment Checklist [86]	Book	Kumar, A., Puranik, M., & Sowmya, K. (2016). Association between dental students’ emotional intelligence and academic performance: a study at six dental colleges in India. Journal of Dental Education, 80(5), 526–532. https://onlinelibrary.wiley.com/doi/pdf/10.1002/j.0022-0337.2016.80.5.tb06112.x?casa_token=aOMTSUUCCjoAAAAA:mfvATJkkTpdQjoGxY2hGU7eUjs3yxzK0rST_ldjQXj_6S0cT6oeQojYJDtcm30dzUx3n8wEOKtBFDJFu (accessed on 8 July 2021)	Unknown	Dimensions: 6 Items: 30
Emotional Intelligence Scale (EIS) [87]	Book	Singh, S., Mohan, M., & Kumar, R. (2011). Enhancing physical health, psychological health and emotional intelligence through Sahaj Marg Raj yoga meditation practice. Indian Journal of Psychological Science, 2, 89–98. http://www.napsindia.org/wp-content/uploads/2017/05/89-98.pdf (accessed on 8 July 2021)	Unknown	Dimensions: 10 Items: 34
Test of Emotional Intelligence (TEMINT) [88]	Paper presented at a congress	Janke, K., Driessen, M., Behnia, B., Wingenfeld, K., & Roepke, S. (2018). Emotional intelligence in patients with posttraumatic stress disorder, borderline personality disorder and healthy controls. Psychiatry Research, 264, 290–296. https://doi.org/10.1016/j.psychres.2018.03.078 (accessed on 8 July 2021)	Ability	Dimensions: unknown Items: 12
Emotional Intelligence Scale—Faces (SIE-T) [89]	Paper of a psychological test laboratory	Piekarska, J. (2020). Determinants of perceived stress in adolescence: the role of personality traits, emotional abilities, trait emotional intelligence, self-efficacy, and self-esteem. Advances in Cognitive Psychology, 16(4), 309. https://doi.org/10.5709/acp-0305-z (accessed on 8 July 2021)	Ability	Dimensions: unknown Items: 18
Test Rozumienia Emocji (TRE) [90]	Peer review article	Piekarska, J. (2020). Determinants of perceived stress in adolescence: the role of personality traits, emotional abilities, trait emotional intelligence, self-efficacy, and self-esteem. Advances in Cognitive Psychology, 16(4), 309. https://doi.org/10.5709/acp-0305-z (accessed on 9 July 2021)	Ability	Dimensions: 5 Items: 30
Emotional Intelligence Index [91]	Peer review article	Veltro, F., Latte, G., Ialenti, V., Bonanni, E., di Padua, P., & Gigantesco, A. (2020). Effectiveness of psycho-educational intervention to promote mental health focused on emotional intelligence in middle-school. Annali dell’Istituto Superiore di Sanità, 56(1), 66–71. https://doi.org/10.4415/ANN_20_01_10 (accessed on 9 July 2021)	Ability	Dimensions: unknown Items: 15
Quick Emotional Intelligence Self-Assessment [92]	Peer review article	https://neotecouncil.org/wp-content/uploads/2012/04/EmotionalIntelligence-Self-Assessment.pdf (accessed on 9 July 2021)	Unknown	Dimensions: 4 Items: 10
Emotional Maturity Scale [93]	Book	Ishfaq, N. & Kamal, A. (2018). Translation and validation of Emotional Maturity Scale on juvenile delinquents of Pakistan. Psycho-Lingua, 48(2), 140–148. https://www.researchgate.net/profile/Nimrah-Ishfaq/publication/334706863_TRANSLATION_AND_VALIDATION_OF_EMOTIONAL_MATURITY_SCALE_ON_JUVENILE_DELINQUENTS_OF_PAKISTAN/links/5d3b01cf4585153e5923c009/TRANSLATION-AND-VALIDATION-OF-EMOTIONAL-MATURITY-SCALE-ON-JUVENILE-DELINQUENTS-OF-PAKISTAN.pdf (accessed on 9 July 2021)	Unknown	Dimensions: 5 Items: 48

## Data Availability

Not applicable.

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
