# Peer review of "Emotional Intelligence Measures: A Systematic Review"

_healthcare, 2021, doi:10.3390/healthcare9121696_

Round 1

Reviewer 1 Report

This is a very interesting study, which provides a valuable contribution both for research and practice. The manuscript is well and clearly written. The method is clearly described and is adequate, considering the goals of the study. The study results are also thoroughly and clearly described. 

Notwithstanding the qualities of the study, there are a few problematic issues and less clear aspects that the authors should address in order to improve the manuscript:

1) The justification for selecting the Web of Science (WoS) database to carry out the bibliographic research is clear. However, the authors to not explain why they did not also use the EBSCO and Scopus databases. Despite their potential overlap with WoS, these databases could yield additional records not contemplated by the WoS.

2) Relatedly, additional records are often identified through other sources (e.g., reference list of records identified through database searching). Did the authors identify any additional records through any kind of hand search?

3) The authors describe their inclusion and exclusion criteria, but do not specify which system was used to define those criteria. Considering the goals of the study, authors could use the SPIDER tool (Cooke et al., 2012) to define and categorized those criteria 

4) The lack of an inter-rater agreement performance is a major limitation of the manuscript. The authors acknowledge this in the discussion, but it is not clear why this methodological place was not conducted. 

5) The authors mention a figure 1, but it is not included in the pdf document for review. A flowchart depicting the several selection phases would be useful: number of records identified through database searching; additional records identified through other sources; records after duplicates removed; records screened and inter-rater agreement; records excluded based on title and abstract; full-text articles assessed for eligibility; full-text articles excluded; articles included in the study.

6) Table 1: a left alignment of the table contents would be a better presentation format and make it easier for the reader to follow through the information presented. Also, a line space between the reliability (i.e., interna consistency, test-retest) and validity (i.e., internal structure, convergent, etc.) would make the information easier to grasp. 

7) The results description in text should not duplicate the information presented in the table (e.g., number of items, alpha coefficients), but should instead present a summary of the detailed information already presented in the table.

Author Response

RESPONSE TO REVIEWER 1:

Review 1: This is a very interesting study, which provides a valuable contribution both for research and practice. The manuscript is well and clearly written. The method is clearly described and is adequate, considering the goals of the study. The study results are also thoroughly and clearly described. Notwithstanding the qualities of the study, there are a few problematic issues and less clear aspects that the authors should address in order to improve the manuscript.

Authors: We thank you very much for the kind words you dedicate to the manuscript, as well as the evaluations you have made of the contributions of the study and of each of the sections. We will keep in mind and work on each of the issues you raise, in order to improve our manuscript.

Point 1: The justification for selecting the Web of Science (WoS) database to carry out the bibliographic research is clear. However, the authors to not explain why they did not also use the EBSCO and Scopus databases. Despite their potential overlap with WoS, these databases could yield additional records not contemplated by the WoS.

Response 1: As you mentioned, we have chosen this database because of the many advantages it has over others, as described in the text. They have not been used anymore, like EBSCO and Scopus, despite their great quality and usefulness, due to the large number of articles that WoS has provided (n = 6,747). However, for future research it would be useful to search these databases. This has been added to the text:

“In future research, it would be convenient to perform the same search in other databases such as EBSCO and Scopus in order to obtain possible articles not contemplated in WoS.”

Point 2: Relatedly, additional records are often identified through other sources (e.g., reference list of records identified through database searching). Did the authors identify any additional records through any kind of hand search?

Response 2: We have not carried out any hand search due to the large number of articles obtained through the WoS search. However, due to the importance of what you propose, we have added it as another aspect to improve in the future:

“A manual search for additional articles, for example, in the references of other articles or in the grey literature, would also be useful.”

Point 3: The authors describe their inclusion and exclusion criteria, but do not specify which system was used to define those criteria. Considering the goals of the study, authors could use the SPIDER tool (Cooke et al., 2012) to define and categorized those criteria.

Response 3: Although initially this great tool that you propose was not used, the inclusion and exclusion criteria take into account part of the components that it uses. However, since our research is not oriented towards the studies themselves (sample, phenomena of interest/interventions, design, etc.), but rather towards the extraction and analysis of the instruments used in these articles, it was not exposed in the methodology. However, we believe that it is a tool with great potential value that can be of great help as an alternative to the PICO tool in the search and analysis of qualitative and mixed studies, and that we will of course use in future research.

Point 4: The lack of an inter-rater agreement performance is a major limitation of the manuscript. The authors acknowledge this in the discussion, but it is not clear why this methodological place was not conducted.

Response 4: Due to logistical and functionality constraints of the research team, the reproducibility of the choice decisions could not be completed in in time. Although a subsequent informal assessment by the research team of the eligibility of the studies showed small differences, these were not important. In addition, we have added this as an aspect to be improved in the future:

“Another very important aspect to be taken into account and to be improved in future research is the performance of a formal evaluation of the inter-rater agreement.”

Point 5: The authors mention a figure 1, but it is not included in the pdf document for review. A flowchart depicting the several selection phases would be useful: number of records identified through database searching; additional records identified through other sources; records after duplicates removed; records screened and inter-rater agreement; records excluded based on title and abstract; full-text articles assessed for eligibility; full-text articles excluded; articles included in the study.

Response 5: We are very sorry that you could not access Figure 1. The manuscript that was uploaded to the journal to be revised contained this figure with all the phases that you propose, but there must have been some problem that made it disappear from the text. We are going to try that this does not happen again in the new update.

Point 6: Table 1: a left alignment of the table contents would be a better presentation format and make it easier for the reader to follow through the information presented. Also, a line space between the reliability (i.e., interna consistency, test-retest) and validity (i.e., internal structure, convergent, etc.) would make the information easier to grasp.

Response 6: Although we initially followed the template provided in the journal, we have taken into account your suggestion regarding alignment and spacing, as you can see in Table 1.

Point 7: The results description in text should not duplicate the information presented in the table (e.g., number of items, alpha coefficients), but should instead present a summary of the detailed information already presented in the table.

Response 7: Thank you for your contribution, we have rewritten the results section so that it presents a summary of the detailed information in the table.

Reviewer 2 Report

First, I would like to say that I am very thankful to have the opportunity to read this study. The suggestions given in this document are intended to improve your work.

General comment:

Although the authors demonstrate knowledge of the subject, the methodology appears to be flawed, compromising the integrity of the work. Unfortunately, I must recommend major revision due to major issues in this manuscript.

Authors are encouraged to read:

  • https://www.bmj.com/content/372/bmj.n71
  • https://www.sciencedirect.com/science/article/pii/S0300893221002748?via%3Dihub
  • (key documents) http://www.prisma-statement.org

This applies to all sections of the document.

Abstract section:

  • The abstract covers almost only the background. As methods the authors indicate the database they have use and the number of tools they have found (results). The abstract must be redone to clearly incorporate introduction, methods, results, and conclusions, as indicated in the journal's guidelines.

 Introduction section:

  • I think the introduction provides interesting information and is well structured, perhaps adding sub-sections would help to make it more organised.
  • All acronyms must be explained, e.g., lines 111.

 Methods section:

  • I am confused by the type of review the authors claim to have carried out. In the title it is called a systematic review, in the abstract a bibliographic review, and in the objectives a ¿descriptive? systematic description.
  • I would like the authors to provide the PRISMA checklist, I do not see sufficient evidence that the study followed the statement.
  • I don't understand the protocol section, have the authors registered the protocol they are going to follow in the review, as indicated in PRISMA, Prospero or any other repository?
  • Did it take 5 months to search a single database?
  • A systematic review cannot be conducted by searching only in one database.
  • The authors seem to have duplicated the eligibility criteria. There is one paragraph (155-162) where this issue is addressed but surprisingly then there is another section (163-173). It should be restructured because it is difficult to understand.
  • Where is the PRISMA diagram?

Author Response

RESPONSE TO REVIEWER 2:

Reviewer 2: First, I would like to say that I am very thankful to have the opportunity to read this study. The suggestions given in this document are intended to improve your work. Although the authors demonstrate knowledge of the subject, the methodology appears to be flawed, compromising the integrity of the work. Unfortunately, I must recommend major revision due to major issues in this manuscript.

Authors: We really appreciate your work reviewing this study. First of all, we would like to thank you for all your comments. They are truly considered useful for us in order to grow in our research careers.

Point 1: Authors are encouraged to read:

  • https://www.bmj.com/content/372/bmj.n71
  • https://www.sciencedirect.com/science/article/pii/S0300893221002748?via%3Dihub
  • https://www.sciencedirect.com/science/article/pii/S0300893221002748?via%3Dihub#tbl0020
  • (Key documents) http://www.prisma-statement.org

This applies to all sections of the document.

Response 1: Thank you very much for your recommendation. All the articles have been reviewed and have provided great information that has been taken into account when adapting and updating our manuscript.

Point 2: The abstract covers almost only the background. As methods the authors indicate the database they have use and the number of tools they have found (results). The abstract must be redone to clearly incorporate introduction, methods, results, and conclusions, as indicated in the journal's guidelines.

Response 2: Thank you for your constructive words. The summary has been modified according to them and to the documents that you have previously provided. However, it has not been possible to go into great depth and comply with the PRISMA checklist for abstracts due to the maximum length required by the journal.

Point 3: I think the introduction provides interesting information and is well structured, perhaps adding sub-sections would help to make it more organised. All acronyms must be explained, e.g., lines 111.

Response 3: As you have indicated, new subsections have been added that allow a clearer reading. In addition, we have explained all the acronyms that appear for the first time, such as those you have named.

Point 4: I am confused by the type of review the authors claim to have carried out. In the title it is called a systematic review, in the abstract a bibliographic review, and in the objectives a ¿descriptive? systematic description.

Response 4: We are very sorry for the confusion, it is true that the text may give rise to doubts. We have replaced all potentially confusing terms with "systematic review".

Point 5: I would like the authors to provide the PRISMA checklist, I do not see sufficient evidence that the study followed the statement.

Response 5: We have added the PRISMA checklist as Annex 1.

Point 6: I don't understand the protocol section, have the authors registered the protocol they are going to follow in the review, as indicated in PRISMA, Prospero or any other repository?

Response 6: Sorry, the title of the protocol subsection does not refer to what was intended. We have modified it.

Point 7: Did it take 5 months to search a single database?

Response 7: Sorry, it was not well expressed. That time was dedicated to the download and subsequent analysis of all the articles read. We have modified the text.

Point 8: A systematic review cannot be conducted by searching only in one database.

Response 8: The review was carried out only in one database due to the large number of results that the search yielded (n = 6,747), and we believe that it can comprise practically almost all the validated instruments that can be found in the literature to measure emotional intelligence. In addition, in order to cover as many instruments as possible, we have accepted articles in all languages and we have not restricted the search to those only written in English.

Point 9: The authors seem to have duplicated the eligibility criteria. There is one paragraph (155-162) where this issue is addressed but surprisingly then there is another section (163-173). It should be restructured because it is difficult to understand.

Response 9: Sorry, it was not well expressed. The first paragraph you mentioned refers to the inclusion criteria of the articles searched and selected (from which we have extracted the instruments), while the second one refers to the inclusion criteria of the instruments themselves included in this study. As you suggested, we have restructured the text and combined them in the eligibility criteria subsection.

Point 10: Where is the PRISMA diagram?

Response 10: We are very sorry that you could not access Figure 1. The manuscript that was uploaded to the journal to be revised contained this figure with all the phases that you propose, but there must have been some problem that made it disappear from the text. We are going to try that this does not happen again in the new update.

Reviewer 3 Report

The presented manuscript presents a systematic review of the measures of emotional intelligence, using the PRISMA method, both from the perspective of ability-based, trait-based and mixed models of emotional intelligence.

In a systematic review, it is always possible to omit some important records, given the large number of articles reviewed. In this case, it is suggested that the works of Fernández-Berrocal et al. Be reviewed, among which there are various works related to adaptations of the Trait Meta-Mood Scale, by Salovey and Mayer, in which an adaptation of the original model of 48 items to another of 24 questions (TMMS-24) in different languages ​​(Fernández-Berrocal, Extremera and Ramos, 2004) in the prestigious Psychological Reports review. The authors may have used only the original versions, but some have been widely viewed in the international context.

The authors in more than one moment of the article state that a search has been carried out from 1900 to 2020 on the Web of Science (WOS), this is reflected in lines 146 and 147 (and also later) . This is not necessary, rather inappropriate since except for some references from the 90s of the last century, all the others are after the year 2000. The creators of the concept emotional intelligence (Salovey & Mayer, 1990) did it with the Publication of his article in 1990, which is why it makes no sense to start the systematic review in 1900, it should have been carried out after 1990.

Otherwise, the manuscript meets the requirements to be published in the journal when they are modified and includes the suggested aspects, under the opinion of this reviewer.

Author Response

RESPONSE TO REVIEWER 3:

Reviewer 2: The presented manuscript presents a systematic review of the measures of emotional intelligence, using the PRISMA method, both from the perspective of ability-based, trait-based and mixed models of emotional intelligence.

Authors: Thank you very much for your review of our manuscript, we appreciate the time you have spent reading and analyzing it.

Point 1: In a systematic review, it is always possible to omit some important records, given the large number of articles reviewed. In this case, it is suggested that the works of Fernández-Berrocal et al. Be reviewed, among which there are various works related to adaptations of the Trait Meta-Mood Scale, by Salovey and Mayer, in which an adaptation of the original model of 48 items to another of 24 questions (TMMS-24) in different languages ​​(Fernández-Berrocal, Extremera and Ramos, 2004) in the prestigious Psychological Reports review. The authors may have used only the original versions, but some have been widely viewed in the international context.

Response 1: As you say and as stated in the manuscript, we have included only the original versions of each tool. However, given the potential value and extensive use of the TMMS-24 by Fernandez-Berrocal et al. (2004), we have added it to the table as another version of the original TMMS noting that it is widely adapted and used, and cited and referenced it appropriately. It has also been cited in the text.

Point 2: The authors in more than one moment of the article state that a search has been carried out from 1900 to 2020 on the Web of Science (WOS), this is reflected in lines 146 and 147 (and also later). This is not necessary, rather inappropriate since except for some references from the 90s of the last century, all the others are after the year 2000. The creators of the concept emotional intelligence (Salovey & Mayer, 1990) did it with the Publication of his article in 1990, which is why it makes no sense to start the systematic review in 1900, it should have been carried out after 1990.

Response 2: You are absolutely right, a priori a search since 1900 does not make sense when the term "emotional intelligence" was conceptualized later, and when the first instrument was developed in 1995. However, we have included all those articles written since 1900 in case any antecedent was found that attempted to conceptualize or measure this construct. A new search since 1990 would be unfeasible, but we keep your words in mind for further research.

Point 3: Otherwise, the manuscript meets the requirements to be published in the journal when they are modified and includes the suggested aspects, under the opinion of this reviewer.

Response 3: Thank you very much for your valuable contributions.

Round 2

Reviewer 2 Report

The authors have made a great effort in this work and compile information that may be of interest to many professionals. I would recommend a final reading, to correct the exclusion criteria (they cannot be the opposite of the inclusion criteria), to indicate everywhere the type of review it is (I still see things other than systematic), and in the limitations to add what they do not comply with in the prism statement, which are several aspects. The abstract and conclusion could be adjusted more. The abstract should be adjusted to what PRISMA asks for and the conclusions should answer the research question and only include aspects of the results of your work, without including knowledge that you had previously.

Author Response

“Regarding the evaluation of the quality of the articles, since our study does not analyse the studies that employ the EI instruments, but the instruments themselves, the assessment of the internal or external validity of the studies is not applicable to this research.”
“Although no protocol was written or registered prior to the research, the inclusion and exclusion criteria for articles and instruments were prior defined. The search was conducted according to these criteria.”
“There are many aspects of the PRISMA statement that, due to the purpose of our re-search, our study does not include (visible as NA in the table in Annex 1). However, it is necessary to develop a protocol for recording the inclusion and exclusion criteria of the primary studies to prevent bias (e.g., bias in the selection process). There are also some methodological aspects to be improved, such as the lack of methods used to assess the risk of bias in the included studies, the preparation or synthesis of the data, or the certainty in the body of evidence of a result. In future research it is necessary to take into account and develop these aspects in order to improve the replicability and methodological validity of the study, and to facilitate the transparency of the research process.”
